# Liver X Receptor Ligand GAC0001E5 Downregulates Antioxidant Capacity and ERBB2/HER2 Expression in HER2-Positive Breast Cancer Cells

**DOI:** 10.3390/cancers16091651

**Published:** 2024-04-25

**Authors:** Asitha Premaratne, Shinjini Basu, Abhinav Bagchi, Tianyi Zhou, Qin Feng, Chin-Yo Lin

**Affiliations:** Center for Nuclear Receptors and Cell Signaling, Department of Biology and Biochemistry, University of Houston, Houston, TX 77004, USA

**Keywords:** HER2-positive breast cancer, liver X receptor, metabolic reprogramming, glutaminolysis, de novo lipogenesis, oxidative stress, induction of apoptosis

## Abstract

**Simple Summary:**

Metabolic reprogramming in HER2-positive breast cancer is associated with acquired resistance to targeted therapies. Thus, targeting dysregulated metabolic pathways in cancers is highly beneficial. In this study, we examined the anti-tumor effects of an LXR inverse agonist GAC0001E5 (1E5) in HER2-positive breast cancer in vitro. Strikingly, we observed disruption of two major pathways, glutaminolysis and de novo lipogenesis, leading to oxidative stress and cell death. Additionally, we discovered the indirect effects of 1E5 in downregulating HER2 expression through fatty acid synthase (FASN). Moreover, in combinatory treatments of 1E5 and lapatinib, 1E5 potentiates lapatinib’s effects. These findings indicate the importance of 1E5 as a metabolic disruptor in HER2-positive breast cancer.

**Abstract:**

The HER2-positive subtype accounts for approximately one-fifth of all breast cancers. Insensitivity and development of acquired resistance to targeted therapies in some patients contribute to their poor prognosis. HER2 overexpression is associated with metabolic reprogramming, facilitating cancer cell growth and survival. Novel liver X receptor (LXR) ligand GAC0001E5 (1E5) has been shown to inhibit cancer cell proliferation by disrupting glutaminolysis and inducing oxidative stress. In this study, HER2-positive breast cancer cells were treated with 1E5 to determine their potential inhibitory effects and mechanisms of action in HER2-positive breast cancers. Similar to previous observations in other cancer types, 1E5 treatments inhibited LXR activity, expression, and cancer cell proliferation. Expression of fatty acid synthesis genes, including fatty acid synthase (*FASN*), was downregulated following 1E5 treatment, and results from co-treatment experiments with an FASN inhibitor suggest that the same pathway is targeted by 1E5. Treatments with 1E5 disrupted glutaminolysis and resulted in increased oxidative stress. Strikingly, HER2 transcript and protein levels were both significantly downregulated by 1E5. Taken together, these findings indicate the therapeutic potential of targeting HER2 overexpression and associated metabolic reprogramming via the modulation of LXR in HER2-positive breast cancers.

## 1. Introduction

HER2-positive breast cancers represent 20% of all breast cancers [1] and are characterized by elevated expression levels of Erb-B2 receptor tyrosine kinase 2 (ERBB2 or HER2) [2]. Increased HER2 activity induces PI3K/AKT [3] and RAF/MEK [4] signaling pathways, facilitating cancer cell proliferation and survival. The standard of care for HER2-positive breast cancers includes surgical resection and adjuvant trastuzumab combined with chemotherapy as first-line therapy, followed by trastuzumab-emtansine conjugate as second-line therapy [5,6,7]. Additionally, tyrosine kinase inhibitors such as lapatinib are used in combination with other therapeutic agents to treat advanced diseases [8]. Despite clinical advances, HER2-positive-acquired therapeutic resistance, early distant metastasis, and recurrence remain major challenges in treating HER2-positive breast cancers [9]. 

Nuclear receptors are ligand-dependent transcription factors that regulate various cellular functions [10,11]. Liver X receptors (LXRs: LXRα and LXRβ) are nuclear receptors that regulate lipid, cholesterol, and glucose metabolism and immune responses [12]. Synthetic ligands have been developed to treat atherosclerosis, and some have been tested in cancer cells for their anti-tumor activities [12,13,14,15]. However, further development was discontinued for synthetic agonists due to elevated liver and circulating triglyceride levels in test animals [16]. To identify novel LXR ligands with anti-tumor activity, we conducted a focused screen of drug-like small molecules predicted by molecular docking to bind LXR [17]. In cell-based screens, we identified GAC0001E5 (1E5) as a potent pancreatic cancer cell proliferation inhibitor. Treatments with 1E5 decreased LXR target gene expression and significantly reduced LXR protein levels, indicative of its inverse agonist and “degrader” activities [17]. Follow-up metabolomic and transcriptomic studies revealed the disruption of glutaminolysis and increased oxidative stress by 1E5 in pancreatic and estrogen receptor-positive and triple-negative breast cancer cells [18,19]. Metabolic reprogramming is a major cancer hallmark characterized by dysregulated metabolic activity [20]. HER2-positive breast cancers are characterized by reprogrammed glycolysis, tricarboxylic acid (TCA) cycle, pentose phosphate pathway (PPP), and glutamine and fatty acid metabolism pathways [21]. These altered pathways function collaboratively to meet the energy demands and elevated antioxidants and macromolecules cancer cells require. LXRs have been shown to regulate gene expression in many of these mechanisms [22]. Specifically, glutaminolysis, the process by which glutamine is converted to glutamate, also produces TCA cycle intermediates and glutathione (GSH) antioxidants, and contributes to the metabolic resilience of cancer cells. Our recent studies in pancreatic ductal adenocarcinoma (PDAC) have revealed that 1E5 acts as an LXR inverse agonist, disrupting glutaminolysis and increasing oxidative stress [18]. In a follow-up study in HER2-negative breast cancers [MCF-7, MCF7-TamR (Tamoxifen-resistant), and MDA-MB-231], the same metabolic pathway was shown to be inhibited [19]. Additionally, lipogenesis is upregulated in HER2-positive breast cancers and is known to amplify HER2 signaling [23]. Key lipogenesis genes (*SREBP1c*, *ACLY*, *ACC*, *FASN*, and *SCD1*) are directly regulated by LXR, and their expression levels upon 1E5 treatment are shown to be downregulated in breast cancers [19]. We therefore posit that targeting LXR with the novel 1E5 ligand may inhibit HER2-positive breast cancers through the disruption of reprogrammed metabolic pathways. 

## 2. Materials and Methods

### 2.1. Treatments, Cell Lines, and Culture

Novel LXR ligand GAC0001E5 (1E5) was synthesized by Otavachemicals (Concord, ON, Canada). LXR agonist GW3965 was received from Cayman Chemicals (Ann Arbor, MI, USA #10054). Tyrosine kinase inhibitor lapatinib and fatty acid synthase inhibitor C75 were purchased from MedChemExpress [South Brunswick Township, NJ, USA (Lapatinib #HY-50898, C75 #HY-12364)]. AU565, HCC-1954, and SKBR3 cell lines were purchased from the American Type Culture Collection (ATCC) (Manassas, VA, USA). AU565 and HCC-1954 cells were cultured in RPMI 1640 (Gibco, Thermo Fisher Scientific, Waltham, MA, USA #11875085). SKBR3 cells were cultured in DMEM (Gibco, #12430047) containing high glucose and HEPES. All media were supplemented with 10% Fetal Bovine Serum (FBS) (Gibco, #26140079). All cell cultures were maintained in a humidified incubator of 5% CO_2_ at 37 °C.

### 2.2. Cell Proliferation Assay

For MTS assays, cells were seeded in a 96-well plate with densities of 1 × 10^4^ cells/well (AU565), 1 × 10^4^ cells/well (SKBR3), and 5 × 10^3^ cells/well (HCC-1954). After 24 h, cells were treated with desired concentrations of the ligands and incubated for 72 h without changing media. After 72 h, NADH-dependent tetrazolium bromide reduction was measured using an MTS assay as an indirect indication of the viable cell quantity (3-(4,5-dimethylthiazol-2-yl)-2,5-diphenyl tetrazolium bromide, Promega, Madison, WI, USA #G3581). A trypan blue exclusion assay was performed to determine viable cell count. Cells were seeded in 6-well plates in different densities—2 × 10^5^ cells/well (AU565), 2 × 10^5^ cells/well (SKBR3), and 1 × 10^5^ cells/well (HCC-1954). After 24 h, the plates were treated with DMSO (vehicle), GW3965 (10 μM), 1E5 (10 μM), and incubated for 72 h. Viable cells were then counted using a hemocytometer upon staining with 0.4% Trypan Blue (Gibco, 15250061).

### 2.3. Real-Time Quantitative PCR

Cells were seeded in 6-well plates in densities described in Section 2.2. After 24 h, treatments were introduced—DMSO (vehicle), GW3965 (10 μM), and 1E5 (10 μM). Total RNA was extracted after 48 h of incubation using the RNeasy Mini Kit (Qiagen, Germantown, MD, USA, #74104). cDNA was synthesized using 1 µg of total RNA with an iScript cDNA synthesis kit (Bio-Rad, Hercules, CA, USA, #1725035). qPCR reactions were set up to be 10 µL in total volume, including (0.5 µL forward primer, 0.5 µL reverse primer, and 1 µL nuclease-free water (Invitrogen, Thermo Fisher Scientific Inc., Waltham, MA, USA, #AM9906) with 5 µL of SYBR Green (Applied Biosystems, Thermo Fisher Scientific Inc., Waltham, MA, USA, #A25742), and 3 µL of 1:10 diluted cDNA. Reactions were loaded to MicroAmp™ Optical 96-Well GPLE Reaction Plates and quantified using a Biosystems 7500 Fast Real-Time PCR system (Applied Biosystems, Thermo Fisher Scientific Inc., Waltham, MA, USA, #4481192). Primers for the study were ordered from the PrimerQuest™ program, IDT, Coralville, Iowa, USA, and the list is provided in Appendix A.

### 2.4. Western Blot Analysis

Cells were seeded in 6 cm plates with the following densities:AU565 (3 × 10^5^ cells/plate), SKBR3 (3 × 10^5^ cells/plate), and HCC-1954 (1.5 × 10^5^ cells/plate). Treatments were incorporated after 24 h, including DMSO (vehicle), GW3965 (10 μM), 1E5 (10 μM), and C75 (50 μM). Upon 48 h of incubation, the culture media were removed. Subsequently, plates were washed twice with 1X PBS. Scraped cell suspensions were centrifuged at 5000 rpm at 4 °C for 5 min, and the supernatant was removed for each. Required amounts (50–100 μL) of lysis buffer (RIPA supplemented with protease inhibitor [Roche Diagnostics, Indianapolis, IN, USA, #11836170001]) were added to each pellet based on pellet size and incubated at 4 °C for 30 min. Cell lysates were then centrifuged at 13,000 rpm at 4 °C for another 15 min to remove cell debris. Protein quantification was performed using VWR Bradford reagent (VWR, Radnor, PA, USA, #E530). 30 µg of total protein was denatured with 5x Laemmli buffer plus 2-Mercaptoethanol at 95 °C for 10 min. The protein samples were separated using SDS-PAGE gels and blotted to PVDF membranes for Western blot analysis (Thermo Fisher Scientific Inc., Waltham, MA, USA, #88518). Transferred blots were incubated in a blocking buffer (5% skimmed milk with TBST) for 1 h before incubating with the primary antibody. HER2 and FASN antibodies were purchased from Cell Signaling Technology (Danvers, MA, USA, HER2 #4290s, FASN #3180T). LXRβ antibody was purchased from R&D systems (Minneapolis, MN, USA, #PP-K8917), and β-actin from Sigma-Aldrich (St. Louis, MO, USA, #A2228). All primary antibodies were incubated with respective blots overnight and washed thrice with 1X TBST prior to adding the appropriate secondary antibody. Protein signals were developed using Clarity^TM^ Western ECL (Bio-Rad, Hercules, CA, USA, #1705061). Generated chemiluminescence signals were captured by LI-COR Odyssey Fc (LI-COR Biosciences, Lincoln, NE, USA, #OFC-0842). 

### 2.5. IC_50_ Assays

Cells were seeded in 96-well plates as described in Section 2.2 for MTS assays. After the initial 24 h window, a concentration gradient of the selected small molecule was added with complete medium and incubated for an additional 72 h. 

1E5—0.01 µM, 0.1 µM, 1 µM, 5 µM, 10 µM, 100 µM, 1000 µM

Lapatinib—1 nM, 10 nM, 100 nM, 1,000 nM, 5000 nM, 10,000 nM, 100,000 nM

C75—1.56 μM, 3.13 μM, 6.25 μM, 12.5 μM, 25 μM, 50 μM, 100 μM

After 72 h, MTS assays were performed, and the data were used to calculate the IC_50_ values. 

### 2.6. Lapatinib+1E5 Combination Treatment

Lapatinib is a tyrosine kinase inhibitor used to treat advanced HER2-positive breast cancer. To investigate the combinatory effect of novel LXR inverse agonist 1E5 and lapatinib in vitro, cells were seeded in 96-well plates according to densities as described in Section 2.2. After 24 h from seeding, DMSO (vehicle), two concentrations of each ligand as single treatments [1E5 (5 µM), 1E5 (10 µM), lapatinib (50 nM), and lapatinib (100 nM)], and the combination [1E5 (5 µM) and lapatinib (50 nM)] were added as treatments. After 72 h, an MTS assay was performed to measure the relative effect of the treatments on the cell viability of the HER2-positive cell lines. 

### 2.7. FASN Inhibitor Analysis

FASN is a crucial enzyme in the de novo lipogenesis pathway and is responsible for converting acetyl-CoA to palmitate [24]. Small molecule C75 acts as an FASN inhibitor binding to β-ketoacyl domain. To delineate the combinatory effects of 1E5 and C75 in HER2-positive breast cancer, treatments were introduced as vehicle DMSO, single treatments [1E5 (5 µM), 1E5 (10 µM), C75 (25 µM), and C75 (50 µM)], and the combination [1E5 (5 µM), C75 (25 µM)]. Cell viability was determined using an MTS assay after 72 h of incubation with the treatments.

### 2.8. Glutamine Dependency Assay

Cells were seeded in 96-well plates in cell densities described in Section 2.2. After 24 h, DMSO and 1E5 (10 μM) treatments were introduced with two separate media conditions, glutamine stripped and supplemented. The entire setup was incubated for 72 h, and an MTS assay was performed to determine the cell viability status. 

### 2.9. Intracellular GSH/GSSG Ratio

Reduced (GSH) and oxidized (GSSG) glutathione levels were detected using a GSH-GSSG-Glo Assay (Promega, Madison, WI, USA #V6611). Cells were seeded in 96-well plates per described densities in xref Section 2.2 with complete medium. After 24 h, DMSO and 1E5 (10 μM) were added. Upon 48 h of treatment, cells were washed with 1X PBS. Subsequently, 50 μL of total/oxidized glutathione lysis buffer was added, and the setup was shaken for 5 min at RT. 50 μL of luciferin-generating reagent was pipetted to each well and incubated for another 30 min at RT. 100 μL luciferin detection reagent was added to each well to measure luciferin content. After 15 min of incubation at RT, luciferin levels were recorded using the Victor X4 plate reader (PerkinElmer, Waltham, MA, USA). All luciferin signal values were normalized to corresponding cell numbers.

### 2.10. ROS Levels

ROS-Glo H_2_O_2_ Assay kit (Promega, Madison, WI, USA #G8820) was used to measure intracellular ROS levels. HER2-positive breast cancer cells AU565 (1 × 10^4^ cells/well), SKBR3 (1 × 10^4^ cells/well), and HCC-1954 (5 × 10^3^ cells/well) were plated in 96-well plates with complete medium. Post 24 h, DMSO and 1E5 (10 μM) treatments were added. After 42 h, 20 μL of 125 μM H_2_O_2_ substrate buffer was added to each well. After 6 h, 50 µL of H_2_O_2_ substrate plus media solution was mixed with 50 µL of ROS-Glo^TM^ detection solution and incubated for 20 min before measuring chemiluminescence using the Victor X4 plate reader (PerkinElmer, Waltham, MA, USA).

### 2.11. TCGA Clinical Data Analysis

RNA sequencing data of breast cancer tissue were collected from The Cancer Genome Atlas (TCGA) breast cancer (BRCA) project, and normal breast tissue was obtained from the TCGA database using TCGAbiolinks (2.30.0) in R (4.3.1). The TCGAquery_subtype function was used to gather and sort HER2-positive breast cancer patients. Normalized transcripts per million (TPM) counts of specific genes in primary tumor and normal solid tissue were plotted. Welch two sample *t*-test was performed to compare the cohorts using the function t.test in the ggplot2 library. TCGAbiolinks and other packages were obtained from https://bioconductor.org and was accessed on 24 February 2024.

## 3. Results

### 3.1. Liver X Receptor Ligand 1E5 Disrupts HER2-Positive Breast Cancer Cell Viability

Previously published studies of the novel LXR ligand 1E5 demonstrated its potent inhibitory effects in pancreatic and breast cancer cells [18,19]. To test their efficacy in HER2-positive breast cancers, three different cell lines (AU565, SKBR3, and HCC-1954) were treated with 1E5, and effects on cell proliferation were measured by tetrazolium salt reduction (MTS) and trypan blue exclusion assays. DMSO was the vehicle, while synthetic LXR ligand GW3965 (GW) was the positive control. Only AU565 cells showed slight inhibition with high concentrations of GW3965 in MTS assays. In contrast, 1E5 treatment showed significant inhibition in all cell lines compared to vehicle and GW3965 (See Figure 1A). Cells were treated at different concentrations varying from 10 nM to 1 mM to investigate the concentration-dependent effects, and IC_50_ values were calculated for all three cell lines (See Figure 1B). HCC-1954 cells were the most responsive to 1E5 treatment (IC_50_ = 6.4 μM), followed by AU565 (IC_50_ = 7.1 μM) and SKBR3 (IC_50_ = 7.3 μM) cells. Inhibitory effects of 1E5 were further characterized and validated in trypan blue exclusion assays (see Figure 1C). GW3965 treatments showed lower cell counts compared to DMSO for all cell lines, but only the 10 μM treatments in AU565 cells showed a significant difference compared to vehicle control. Similar to the MTS assay results, 1E5 treatments showed significant inhibition in all cell lines (see Figure 1C). These findings provide the first evidence of the effectiveness of targeting LXR with novel ligand 1E5 in HER2-positive breast cancers. 

### 3.2. 1E5 Is an LXR Inverse Agonist in HER2-Positive Breast Cancer Cells

To determine the mechanisms of action of 1E5 in HER2-positive breast cancer, we measured mRNA and protein levels of LXR and selected LXR target genes upon treatment with DMSO, GW (10 μM), and 1E5 (10 μM). The mRNA levels of LXRβ, the most abundant LXR isotype in HER2-positive cell lines, were downregulated following 1E5 treatment in SKBR3 and HCC-1954 cells, while a slight upregulation was observed in AU565 cells (see Figure 2A). LXR agonist GW3965 upregulated LXRβ transcript levels except in SKBR3 cells, where a minor downregulation was observed. LXRβ protein levels were analyzed using Western blot following control and ligand treatments, and LXRβ protein levels were significantly reduced with 1E5 treatment (see Figure 2B) while GW3965 elicited an upregulation (see Figure 2C). To determine the effects of ligand treatments on LXR activity and its target gene expression, mRNA levels of target genes were measured by quantitative PCR following treatments. As expected, synthetic agonist GW3965 upregulated the selected LXR target genes in all cell lines (see Figure 2C). Treatments with 1E5 downregulated the expression of target genes SREBP1c [25], ACC, FASN, and SCD while slightly upregulating ABCA1 [26] and ABCG1. These results indicate that 1E5 inhibits LXR activity and expression as an inverse agonist and ”degrader” in HER2-positive breast cancer cells. 

### 3.3. HER2 Expression Is Downregulated by 1E5 

HER2 is a receptor tyrosine kinase and forms homodimers or heterodimers with HER1, HER3, or HER4. Dimerization upon ligand binding leads to downstream activation of PI3K/AKT [3] and RAF/MEK [4] signaling pathways which drive cell proliferation and survival. Since LXR functions as a ligand-dependent transcription factor, we posited that modulation of its activity and expression may affect the expression of HER2 and other essential growth-regulatory genes. LXR inverse agonist 1E5 downregulated HER2 transcript levels in all three cell lines, while agonist GW3965 downregulated HER2 only in AU565 (see Figure 3A). Treatments with ligands GW3965 and 1E5 reduced HER2 protein levels, with 1E5 treatments showing a markedly more significant reduction (see Figure 3B). All four HER gene family transcript levels were downregulated by 1E5 across all three cell lines, excluding undetectable levels of HER4 in HCC-1954 (see Figure 3C). These results indicate that 1E5 can not only modulate the expression of LXR but also the expression of key growth factor receptors in HER2-positive breast cancers.

### 3.4. Additive Effects of 1E5 and Lapatinib Suggest Complementary Targeting of HER2

Lapatinib is a tyrosine kinase inhibitor that mechanistically blocks ATP binding in the intracellular tyrosine kinase domain of HER2, and hinders the downstream signaling through PI3K/AKT and RAF/MEK pathways. Even though tyrosine kinase inhibitors show major clinical advances in treating HER2-positive breast cancer, de novo and acquired resistance remain major limitations [27]. Combination treatment strategies are currently being used to bypass therapy resistance. To determine potential crosstalk between LXR and HER2 signaling, HER2-positive breast cancer cells were treated with novel LXR ligand 1E5 and lapatinib. Prior to the co-treatment studies, the effective concentrations of lapatinib in the selected cell lines were determined following treatments with varying concentrations of the inhibitor. AU565 was the most sensitive cell line (IC_50_ = 55.8 nM), while SKBR3 (IC_50_ = 119.8 nM) and HCC-1954 (IC_50_ = 530 nM) showed lower sensitivities (see Figure 4A). Treatments with 1E5 (10 μM) and lapatinib (100 nM) showed inhibitory effects. In all three cell lines, the combination of half concentration of 1E5 (5 μM) and lapatinib (50 nM) showed additive effects (see Figure 4B). These results suggest that both compounds target the same mechanism, possibly through inhibitory effects on the expression of HER2 (1E5) and the inhibition of HER2 kinase activity (lapatinib).

### 3.5. Expression of Fatty Acid Synthesis Genes Is Disrupted by 1E5

Upregulated de novo lipogenesis is a component of metabolic reprogramming in HER2-positive breast cancer [28]. The key enzyme fatty acid synthase (FASN) converts acetyl-CoA to palmitate to produce high levels of free fatty acids (FFA). Elevated FASN transcript levels are associated with poor prognosis in clinical samples. HER2 and FASN upregulate each other to create a positive feedback loop to facilitate HER2-mediated tumorigenesis [23,29]. FASN is a known LXR target gene, and treatments with 1E5 downregulated FASN transcript levels as shown in Figure 2D. To further characterize the interaction between LXR, FASN, and HER2, selected cell lines were treated with 1E5 in combination with FASN inhibitor C75 [30]. Treatments with C75 disrupted HER2-positive breast cancer cell proliferation in a concentration-dependent manner (see Figure 5A). Combined treatment with 1E5 and C75 revealed an additive effect in AU565 and SKBR3 (see Figure 5B), suggesting that the same pathway is targeted by both compounds. Furthermore, FASN and HER2 protein levels were significantly decreased following 1E5 and C75 treatments, except in SKBR3 cells, where reduction in FASN was not statistically significant (see Figure 5C,D). LXR agonist GW treatments showed no change in FASN levels but slight reductions in HER2 across all cell lines. These results provide evidence that 1E5 may downregulate HER2 protein expression by disrupting FASN activity and enzyme levels in HER2-positive breast cancer.

### 3.6. LXR Ligand 1E5 Suppresses Glutaminolysis and Induces Oxidative Stress

Upregulated glutaminolysis is a hallmark of several malignancies, including breast cancers [31]. Glutaminase 1 (GLS1) converts glutamine to glutamate and is the rate-limiting enzyme in this pathway. Glutamate is a vital metabolic intermediary of glutaminolysis which regulates the TCA cycle, amino acid biosynthesis, nucleic acid production, and mTOR signaling [32]. To determine the effects of 1E5 on the expression of glutaminolysis genes, mRNA levels were measured using Quantitative PCR following ligand treatments. Expression of the gene-encoding glutaminase1 (GLS1) was downregulated in all cell lines (see Figure 6A). Similar downregulation was observed for glutaminolysis-related genes GOT1, GOT2, and GLUD1. The vulnerability of HER2-positive breast cancer cells to disruption of glutaminolysis was confirmed by removing exogenous glutamine from the culture medium and the subsequent inhibition of cell proliferation (see Figure 6B). Glutathione (GSH) is a tripeptide made up of cysteine, glutamate, and glycine which function as the major antioxidants in cells. Upregulated glutaminolysis in cancers can elevate GSH levels to mitigate oxidative stress resulting from increased metabolic activity. The antioxidant capacity can be measured as a ratio between reduced (GSH) and oxidized (GSSG) glutathione levels. Treatments with 1E5 (10 μM) significantly reduced antioxidant capacity in all cell lines (see Figure 6C). As a result, reactive oxygen species (ROS) in HER2-positive breast cancer significantly increase (see Figure 6D). Although low to medium levels of ROS are beneficial for cancer cell homeostasis, higher spikes of ROS can promote cell death [33]. Similarly, inhibition of lipogenesis by C75 has also been shown to induce apoptosis [34]. Poly(ADP-ribose) polymerase (PARP-1) acts as a DNA repair enzyme, and the cleavage of this enzyme to its 89 kDa and 24 kDa fragments is a hallmark of apoptosis. Treatments with 1E5-induced PARP cleavage across all cell lines (see Figure 7) similar to C75. Interestingly, LXR agonist GW3965 also induced apoptosis in AU565 cells.

### 3.7. Lipogenesis Genes Downregulated by 1E5 Are Overexpressed in HER2-Positive Breast Cancers

Lipogenesis genes are among LXR target genes. Treatments with 1E5 significantly reduced the expression of LXR target genes. To determine the clinical relevance of the affected genes and pathways, their expression was examined in a cohort of HER2-positive breast cancer patients and in normal breast tissue. 

No differences were observed in LXRβ (NR1H2) transcript levels (see Figure 8). Transcript levels of SREBF1/SREBP1c, FASN, ACACA, and ACLY were elevated in patient samples as compared to normal tissues. These results indicate that metabolic reprogramming in HER2-positive cancer cells involves upregulation of genes involved in de novo lipogenesis, and targeting LXR with novel ligand 1E5 is a potential strategy for exploiting metabolic changes in tumor cells.

## 4. Discussion

Previously, there has been an effort to repurpose atherosclerosis-specific LXR agonists GW3965 and T0901317 to target cancer. During the pre-clinical phases, many of these studies displayed complications such as hypertriglyceridemia and neurotoxicity that were shown to outweigh the anti-tumor benefits [16,35]. This substantiates the need for the development of cancer-specific LXR ligands. A few LXR inverse agonists have since emerged, including SR9243, which enhanced anti-tumor immune responses in TNBCs [36,37]. To specifically identify LXR ligands with tumor inhibitory properties, we conducted a screen and identified 1E5 as an LXR inverse agonist and degrader with potent inhibitory activities in cancer cells [17]. In this current study, 1E5 demonstrated significant disruption of cell proliferation in a concentration-dependent manner in all HER2-positive cell lines used (AU565, SKBR3, and HCC-1954). Activation of HER2 and subsequent downstream signaling of PI3K/Akt and Raf/MEK pathways is the major driver of HER2-positive breast cancer proliferation and survival. Strikingly, 1E5 treatments suppressed both HER2 transcript and protein levels in all three cell lines. Additionally, the expressions of related growth factor receptor genes EGFR (HER1), HER3, and HER4 were also downregulated. Currently, the targeted therapeutic strategies in HER2-positive breast cancer include inhibiting HER2 dimerization by monoclonal antibodies like trastuzumab and inhibiting HER2 kinase activity with small molecule inhibitors such as lapatinib. All these strategies interfere with HER2 activity and disrupt downstream signaling to inhibit cancer cell proliferation and survival and promote cell death. Insensitivity and acquired resistance to anti-HER2 monoclonal antibodies and tyrosine kinase inhibitors are major challenges in the treatment of HER2-positive breast cancers. Our findings suggest that targeting HER2 expression through modulation of LXR activity can complement existing treatment strategies and target mechanisms. Dysregulation of fatty acid synthesis is associated with HER2-positive breast cancers. Elevated free fatty acid levels help maintain higher proliferation rates by stabilizing cell membrane and enhancing energetics [24]. FASN is under the direct transcriptional control of LXRβ in breast cancer [19].

Inhibition of FASN function by C75 has been shown to lower HER2 protein levels and activity and promote cell death [38]. Moreover, pharmacological inhibition of FASN in HER2-positive breast cancer cells resulted in the accumulation of ETV4/PEA3, a transcriptional repressor of the *HER2* gene, and reduced HER2 transcript and protein levels [39]. ETV3/PEA3 is a member of the ETS family of DNA-binding proteins, and it has been shown to bind directly to the *HER2* promoter and suppress HER2 expression [40]. In this study, 1E5 treatments reduced both FASN and HER2 transcript (Figure 2D) and protein levels (Figure 5D). It is possible that downregulation of FASN by 1E5 may similarly increase ETV3/PEA3 in HER2-positive breast cancer cells. Combined treatments with 1E5 and C75 resulted in additive inhibitory effects (see Figure 5B), and these findings suggest that 1E5 and C75 may act on the same regulatory mechanism, possibly through the induction of ETV4/PEA3 accumulation, although it is possible that 1E5 may act on other mechanisms which regulate HER2 expression. Future studies will address these possible mechanisms which mediate the downregulation of HER2 transcript and protein levels by 1E5. Treatments with 1E5 disrupted glutaminolysis and reduced glutathione (GSH) levels, thereby promoting ROS accumulation (in this study and previously published studies) of pancreatic cancer cells and other breast cancer cell types [18,19]. The effects of 1E5 on glutaminolysis are also likely to impact fatty acid synthesis [41]. Citrate is a TCA cycle intermediate that acts as a substrate for fatty acid synthesis. It is an intermediate that bridges glutamine metabolism and fatty acid synthesis, and reduced glutaminolysis can thus downregulate de novo lipogenesis. Disruption of these pathways is a promising target mechanism in cancer drug discovery. Glutaminase inhibitor DRP-104 is currently being tested in clinical trials [42]. FASN inhibitor TVB-2640 is specifically being evaluated in the treatment of HER2-positive cancer in combination with paclitaxel and trastuzumab [43]. Findings from this study suggest that HER2 and these critical pathways in cancer metabolism can similarly be disrupted by LXR modulators such as 1E5.

## 5. Conclusions

The work presented in this article represents the potential of 1E5 as a promising metabolic disruptor in vitro. In the future, characterizations of 1E5 in lapatinib and trastuzumab-resistant cell models will be beneficial in delineating the re-sensitization effects. 1E5-induced tumor reduction and pharmacokinetic and pharmacodynamic properties must be evaluated to further assess its potential in pre-clinical models. 

## Figures and Tables

**Figure 1 cancers-16-01651-f001:**
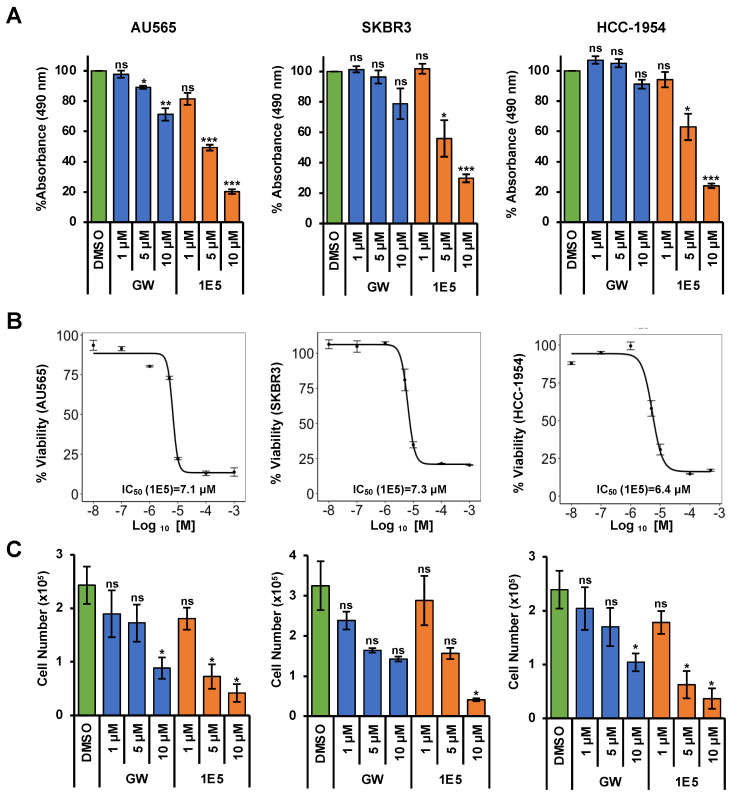
Treatments with 1E5 inhibit HER2-positive breast cancer cell proliferation. (**A**) MTS assays were performed following treatments of AU565, SKBR3, and HCC-1954 cell lines with GW3965 and 1E5 ligand incubated for 72 h at three different concentrations (1 µM, 5 µM, and 10 µM). (**B**) IC_50_ calculations for 1E5 in HER2-positive breast cancer cell lines using MTS assay data after 72 h of treatments with increasing concentrations (0.01 µM, 0.1 μM, 1 μM, 5 μM, 10 μM, 100 μM, and 1000 μM) of 1E5. (**C**) Treatment results were validated by cell viability counts utilizing trypan blue exclusion assays following treatments of GW and 1E5 (1 µM, 5 µM, and 10 µM). All the data represent three biological replicates (*n* = 3). Standard error was plotted in the graphs as necessary. Student’s *t*-test was used to calculate statistical significance (two-tailed, two-sample equal variance), where significance is indicated by * *p* < 0.05, ** *p* < 0.01, *** *p* < 0.001, and not significant (ns) *p* > 0.05.

**Figure 2 cancers-16-01651-f002:**
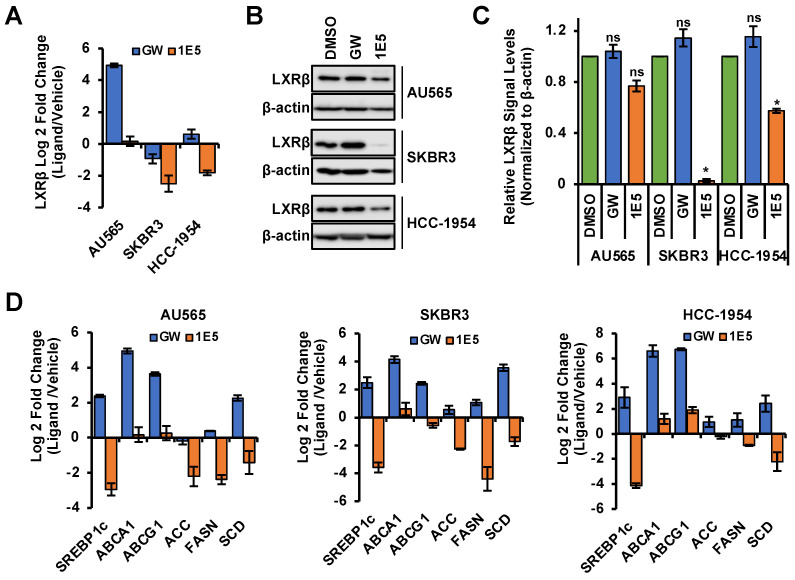
Novel ligand 1E5 is an LXR inverse agonist and degrader which downregulates activity and protein levels. (**A**) LXRβ transcript levels were disrupted with 1E5 (10 μM) treatment after 48 h. (**B**,**C**) LXRβ protein levels significantly reduce with 1E5 (10 μM) and slightly upregulate with GW3965 (10 μM) following 72 h of treatment. (**D**) qPCR data for LXR target genes upon 48 h of GW and 1E5 (10 μM) treatments. Genes involved in de novo lipogenesis SREBP1c, ACC, FASN, and SCD1 were downregulated upon 1E5 treatment. Data were obtained from three biological replicates (*n* = 3). Standard error was plotted in the graphs as necessary. Student’s *t*-test was used to calculate statistical significance (two-tailed, two-sample equal variance), where * *p* < 0.05, and not significant (ns) *p* > 0.05.

**Figure 3 cancers-16-01651-f003:**
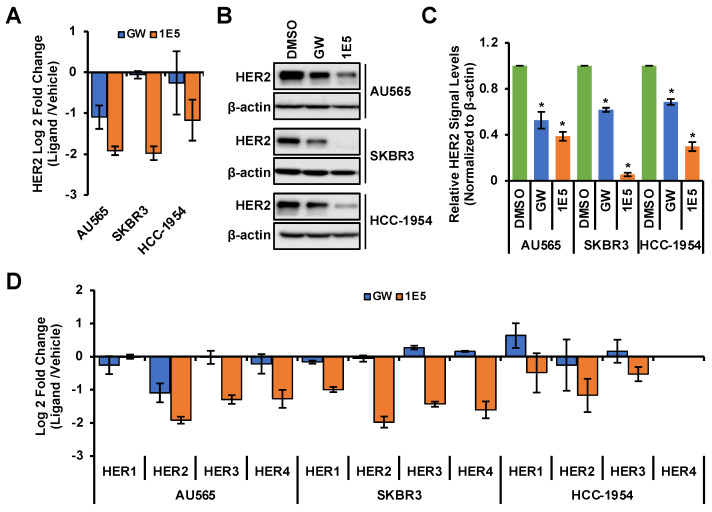
The expression of HER2 and other family members was downregulated by 1E5 treatment. (**A**–**C**) HER2 transcript and protein levels were measured following treatments with 1E5 (10 µM) and GW (10 μM) for 72 h. (**D**) Transcript levels for EGFR (HER1), HER3, and HER4 were determined following 48 h of 1E5 (10 μM) treatment. HER4 levels in HCC-1954 were undetectable. Data were collected from three biological replicates (*n* = 3). Standard error was plotted in the graphs. Student’s *t*-test was used to calculate statistical significance (two-tailed, two-sample equal variance), where * *p* < 0.05.

**Figure 4 cancers-16-01651-f004:**
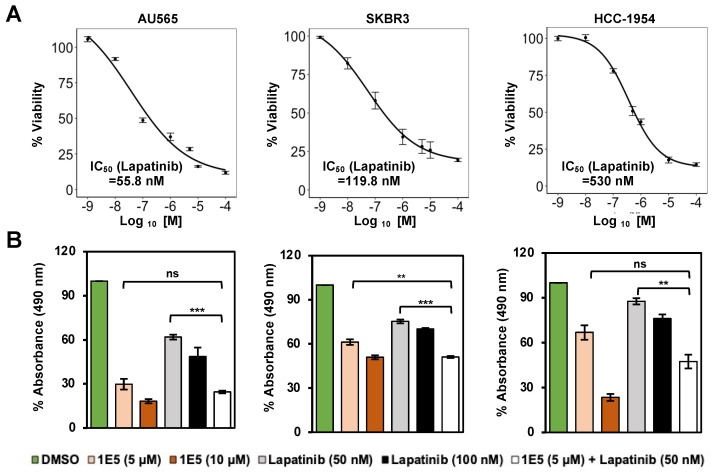
Co-treatment with 1E5 and lapatinib revealed additive inhibitory effects. (**A**) Lapatinib IC_50_ was measured following treatments with varying concentrations (1 nM to 100,000 nM). (**B**) Combination treatments with 1E5 (5 μM) and lapatinib (50 nM) exhibited additive effects. The data were collected from three biological replicates (*n* = 3). Standard error was plotted in the graphs. Student’s *t*-test was used to calculate statistical significance (two-tailed, two-sample equal variance), where ** *p* < 0.01, *** *p* < 0.001, and not significant (ns) *p* > 0.05.

**Figure 5 cancers-16-01651-f005:**
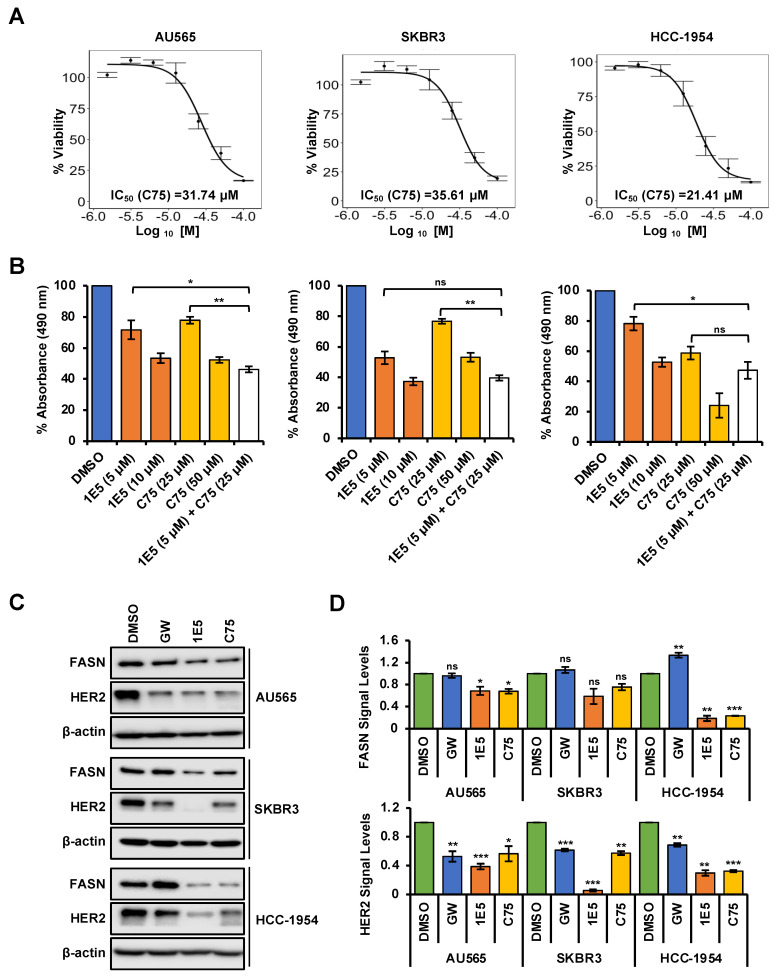
Crosstalk between LXR and FASN can disrupt HER2 expression. (**A**) The effect of C75 in HER2-positive breast cancer cells was analyzed after 72 h of incubation with a concentration gradient (0.625 μM to 100 μM) followed by MTS assays. IC_50_ calculations were performed using the resulting data. (**B**) Combination treatments showed additive effects, except in HCC-1954. (**C**,**D**) 1E5 and C75 treatments resulted in downregulating FASN and HER2 protein levels, except in the SKBR3 cell line, where only HER2 levels reduced. Additionally, GW-mediated disruption was only observed with HER2, whereas FASN showed no significant upregulation. The data represent three biological replicates (*n* = 3). Standard error was plotted as error bars in the graphs. Student’s *t*-test was used to calculate statistical significance (two-tailed, two-sample equal variance), where * *p* < 0.05, ** *p* < 0.01, *** *p* < 0.001, and not significant (ns) *p* > 0.05.

**Figure 6 cancers-16-01651-f006:**
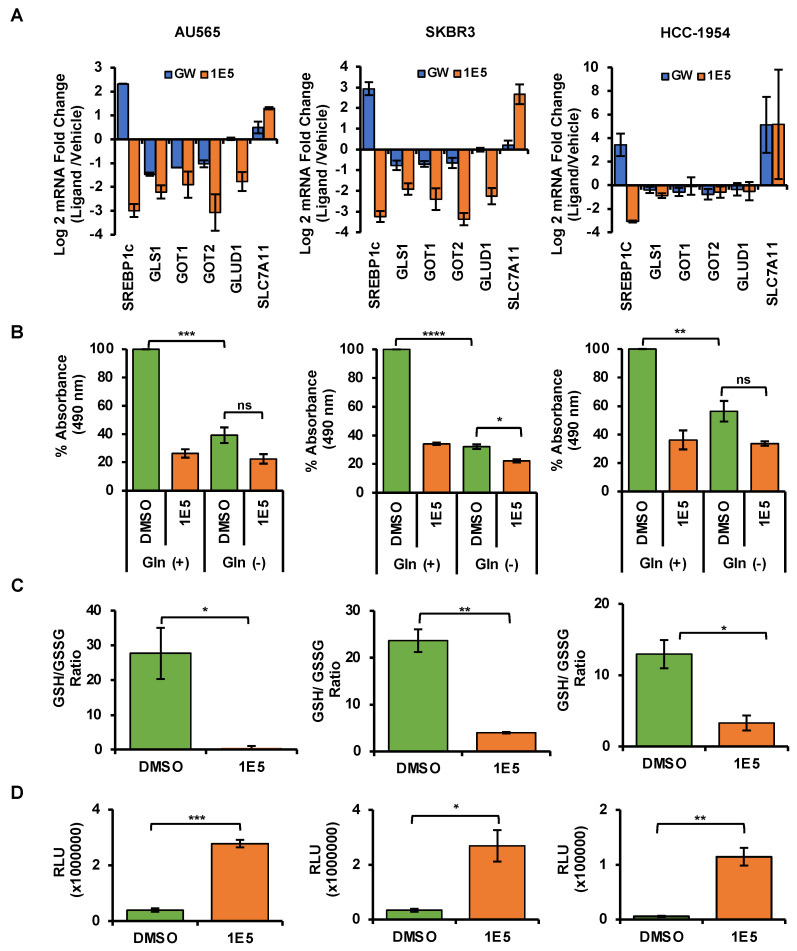
Treatments with 1E5 inhibit glutaminolysis and induce oxidative stress. (**A**) Glutaminolysis-related genes were downregulated after 48 h of 1E5 (10 μM) treatments. (**B**) When exogenous glutamine was removed, low levels of cell viability were observed, indicating the glutamine dependence of selected cell lines. (**C**,**D**) Higher ratios of reduced (GSH) to oxidized (GSSG) glutathione are indicators of antioxidant capacity and were confirmed by increases in ROS. Data were collected in three biological replicates (*n* = 3). Standard error was indicated as error bars in the graphs. Student’s *t*-test was used to calculate statistical significance (two-tailed, two-sample equal variance), where * *p* < 0.05, ** *p* < 0.01, *** *p* < 0.001, **** *p* < 0.0001, and not significant (ns) *p* > 0.05.

**Figure 7 cancers-16-01651-f007:**
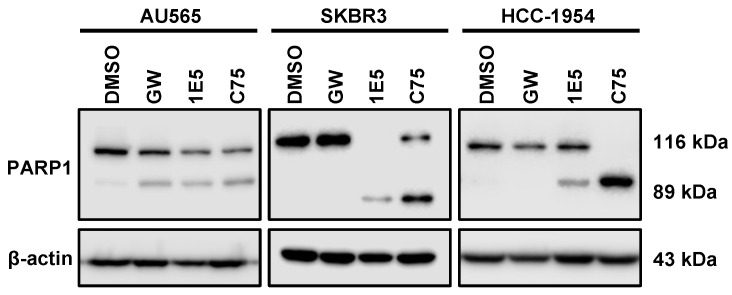
PARP cleavage is induced by 1E5 in HER2-positive breast cancer cells. Enzymatic cleavage of Poly(ADP-ribose) polymerase 1 (PARP-1) is a hallmark of apoptosis. Cleaved products 116 kDa and 89 kDa were observed in C75 (30 μM) and 1E5 (10 μM) treated samples across all cell lines. Interestingly, agonist GW3965 induced apoptosis only in the AU565 cell line. All the data here represent three biological replicates (*n* = 3).

**Figure 8 cancers-16-01651-f008:**
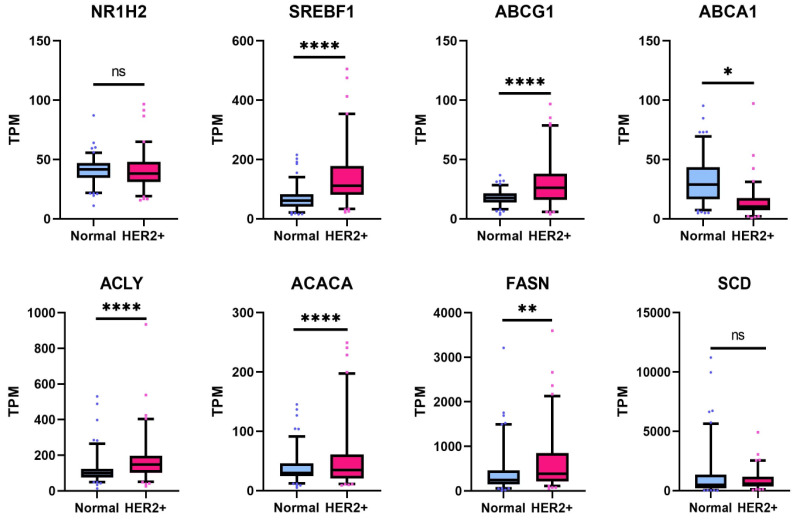
LXR target gene transcript levels are upregulated in the HER2-positive breast cancer patient cohort. Expression of LXR target genes in HER2-positive tumors was compared to normal breast tissue, and statistical significance was determined by Welch’s two-sample *t*-test, where * *p* < 0.05, ** *p* < 0.01, **** *p* < 0.0001, and not significant (ns) *p* > 0.05.

## Data Availability

TCGA data presented in these studies are available on publicly available databases.

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
