# Peer review of "Liver X Receptor Ligand GAC0001E5 Downregulates Antioxidant Capacity and ERBB2/HER2 Expression in HER2-Positive Breast Cancer Cells"

_cancers, 2024, doi:10.3390/cancers16091651_

Round 1
Reviewer 1 Report
Comments and Suggestions for Authors
The authors examined the potential of novel LXR inverse agonist 1E5 as an anti-cancer drug in HER2-positive breast cancers.
Although some data are interesting, there are still some points be addressed.
It is confusing that both LXR inverse agonist (1E5) and agonist (GW3695) inhibits the proliferation and HER2 expression. Are these effects independent of LXR?
The authors stated that 1E5 may downregulate HER2 protein expression by disrupting FASN activity and 327 enzyme levels in HER2-positive breast cancer. Please provide more information or explanation. How does 1E5 disrupt FASN activity? Does 1E5 binds to FASN (in LXR-dependent or independent manner)?
The authors demonstrated that combined effect of 1E5 and lapatinib, or 1E5 and C75 on HER2-positive breast cancer cells and concluded that same pathway is targeted by both compounds. However, it is confusing. Additive effect may be obtained even if these compounds target different pathway.
Author Response
The authors examined the potential of novel LXR inverse agonist 1E5 as an anti-cancer drug in HER2-positive breast cancers.
Although some data are interesting, there are still some points be addressed.
- It is confusing that both LXR inverse agonist (1E5) and agonist (GW3695) inhibits the proliferation and HER2 expression. Are these effects independent of LXR?
Thank you for highlighting these interesting and somewhat contradictory observations from our study and for posing an important question regarding the mechanisms of action of LXR ligands. We have provided the following explanations and also made appropriate revisions in the manuscript to address these comments:
While both GW3965 and 1E5 showed inhibitory effects, 1E5 showed greater effects on cancer cell viability as compared to GW3965, and we believe that their mechanisms of inhibition are distinct. In other breast cancer cell models, we have shown that GW3965 downregulates the expression of E2F transcription factors which regulate the expression of genes involved in cell cycle progression (Nguyen-Vu et al., 2013). On the other hand, in these and our previous studies in ER+ and triple-negative breast cancer cells (Premaratne et al., 2023),1E5 downregulates glutaminolysis and thereby decreases glutathione levels, leading to a corresponding increase in reactive oxygen species (ROS) and oxidative stress (Figure 6). It appears that hyperactivation by the addition of synthetic agonists or inhibition by inverse agonists of LXR can lead to homeostatic imbalance within the cancer cells and thereby decrease cell proliferation and survival.
Similarly, HER2 receptor levels are downregulated by both GW3965 and 1E5 treatments but more drastically with 1E5 (Figure 3B and C) where HER2 protein levels are greatly reduced or not detectable following treatments. One possible mechanism for the 1E5-driven HER2 downregulation is through its downregulation of the LXR target gene FASN which encodes fatty acid synthase. Pharmacological inhibition of FASN using cerulenin and C75 has been demonstrated to induce the accumulation of the transcription factor ETV4/PEA3, a transcriptional repressor of HER2 and subsequent downregulation of HER2 expression (Ventura et al., 2015; Menendez et al., 2004; Xing et al., 2000). In this study, 1E5 treatment downregulates FASN transcript (Figure 2D) and protein (Figure 5D) levels and also HER2 transcript and protein (Figure 3) levels as well. These results provide evidence that ETV4/PEA3 may be involved in mediating the effects on HER2 expression. For the downregulation of HER2 by GW3965, the relatively modest deceases in HER2 protein levels in all three cell lines are not associated with similar decreases in transcript levels in two of the cell lines. It is likely that the mechanisms of HER2 downregulation differ between the two ligands.
We have previously sought to address whether the effects of ligands are dependent on LXR. To do so, we designed gene knockdown experiments, with the intent of characterizing the response of cancer cells to ligand treatments following the knockdown of LXR expression. In the preliminary studies to determine the efficacy of the siRNAs, prior to conducting the ligand treatments, we observed a significant decrease in cell viability following knockdown. Due to this, we were not able to carry out the treatments to more definitively address the question of whether response to ligand was dependent on LXR. The published knockdown results (Candelaria et al., 2014), however, highlight a critical role for LXR in cancer cell proliferation and survival, and these findings are consistent with the inhibitory effects of 1E5 and the reduction in LXR protein levels following treatment.
We have included these explanations prompted by the reviewer’s comments in the revised Discussion section (lines 428 - 442) and have highlighted them in yellow.
- The authors stated that 1E5 may downregulate HER2 protein expression by disrupting FASN activity and enzyme levels in HER2-positive breast cancer. Please provide more information or explanation. How does 1E5 disrupt FASN activity? Does 1E5 binds to FASN (in LXR-dependent or independent manner)?
Please see a more detailed explanation provided in response to the previous comment. Briefly, 1E5 inhibits LXR activity and reduces LXR protein levels, leading to decreased FASN expression. Therefore, the effects of 1E5 on FASN is through its modulation of LXR activity and expression. We hypothesize that this reduction in FASN expression and overall activity likely lead to accumulation of ETV4/PEA3, a transcriptional repressor of the HER2 gene.
- The authors demonstrated that combined effect of 1E5 and lapatinib, or 1E5 and C75 on HER2-positive breast cancer cells and concluded that same pathway is targeted by both compounds. However, it is confusing. Additive effect may be obtained even if these compounds target different pathway.
We appreciate the point made here that additive effects may be obtained even if the compounds target different pathways, but provide an explanation below regarding our conclusion that they likely target the same pathway:
Regarding the combinations used in this study, lapatinib acts as a tyrosine kinase inhibitor, blocking HER2 activation via tyrosine phosphorylation and downstream PI3K/AKT and REF/MEK signaling. On the other hand, 1E5 is associated with disruption of HER2 transcript and protein expression, presumably through ETV4/PEA3 accumulation due to low levels of FASN. Thus, one possible mechanism of additive effects observed can be due to 1E5 and lapatinib targeting different nodes of the HER2 signaling pathway. Relatedly, C75 is a pharmacological inhibitor of FASN, a key enzyme in de novo lipogenesis. 1E5 and C75 were shown in this study to disrupt FASN and HER2 protein levels in most cell lines (Figure 5D). Given the FASN disruption caused by these ligands, it’s reasonable to conclude that they both act on HER2 expression. We do agree with the reviewer, however, that there can be other mechanisms at play. As indicated previously, these explanations are included in the revised manuscript (lines 428-442).
Once again, we thank the reviewer for the careful review and thoughtful comments and recommendations. We believe that our revisions based on the comments have improved the manuscript.
Reviewer 2 Report
Comments and Suggestions for Authors
In this study, the authors aimed to study the effects and mechanisms of the novel LXR ligand 1E5 in HER2-positive breast cancer cells. They found that 1E5 inhibited cancer cell proliferation by disrupting glutaminolysis and inducing oxidative stress. It also downregulated FASN and HER2 levels, suggesting a potential therapeutic strategy for HER2-positive breast cancer. Their findings provide valuable insights into the therapeutic potential of targeting HER2 overexpression and associated metabolic reprogramming through LXR modulation in HER2-positive breast cancers.
Major criticizes:
1) To strengthen the main point that targeting LXR with the novel 1E5 ligand may inhibit HER2-positive breast cancers through the disruption of reprogrammed metabolic pathways, it is important to include a HER2-negative as control for comparison.
2) Figure 1C: In trypan blue exclusion assays, treatment with 5μM and 10μM GW in SKBR3 and 10μM GW in HCC-1954 cells is unlikely to be nonsignificant compared to the vehicle control, as indicated by the bar graph results. I also doubt the absorbance results of GW treatment because the difference between cell number and absorbance within the same cell and treatment is too large. Please carefully redo all the statistical analysis.
Comments on the Quality of English Language1) Line 17: “HER-positive breast cancer.” Should be “HER2-positive breast cancer.”
2) Line 73-75: “Key lipogenesis genes (SREBP1c, ACLY, ACC, FASN, and SCD1) are directly regulated by LXR, and their expression levels upon 1E5 treatment.” This phrase is incomplete.
3) Line 237: “3.2.1. E5 is an LXR Inverse Agonist in HER2-positive Breast Cancer Cells” should be “3.2. 1E5 is an LXR Inverse Agonist in HER2-positive Breast Cancer Cells”
Author Response
In this study, the authors aimed to study the effects and mechanisms of the novel LXR ligand 1E5 in HER2-positive breast cancer cells. They found that 1E5 inhibited cancer cell proliferation by disrupting glutaminolysis and inducing oxidative stress. It also downregulated FASN and HER2 levels, suggesting a potential therapeutic strategy for HER2-positive breast cancer. Their findings provide valuable insights into the therapeutic potential of targeting HER2 overexpression and associated metabolic reprogramming through LXR modulation in HER2-positive breast cancers.
Major criticizes:
- To strengthen the main point that targeting LXR with the novel 1E5 ligand may inhibit HER2-positive breast cancers through the disruption of reprogrammed metabolic pathways, it is important to include a HER2-negative as control for comparison.
We agree with the reviewer and apologize for not providing the appropriate background information and our previously published findings in HER2-negative breast cancer cell lines for comparison (Premaratne et al., 2023). In our first set of studies characterizing the effect of 1E5 in breast cancer, three major cell lines were used [MCF7, MCF7-TamR (tamoxifen-resistant), and MDA-MB-231]. The key findings in these HER2-negative cells were that 1E5 acts as an LXR inverse agonist and LXR degrader and disrupts glutaminolysis and corresponding glutathione production, leading to the accumulation of reactive oxygen species. In comparison, 1E5 treatment of HER2-positive breast cancer cells in the current study have similar effects and target the same metabolic pathways. Strikingly, in HER2-positive cells, we discovered that 1E5 can inhibit HER2 transcript and receptor protein expression, likely through FASN downregulation and posited accumulation of a transcriptional repressor ETV4/PEA3. We’ve provided a more detailed description of the published HER2-negative studies in the revised Introduction section (Lines 70 - 74) and the novel mechanisms involving FASN and possibly ETV4/PEA3 has been added to the Discussion section (lines 330-339).
- Figure 1C: In trypan blue exclusion assays, treatment with 5μM and 10μM GW in SKBR3 and 10μM GW in HCC-1954 cells is unlikely to be nonsignificant compared to the vehicle control, as indicated by the bar graph results. I also doubt the absorbance results of GW treatment because the difference between cell number and absorbance within the same cell and treatment is too large. Please carefully redo all the statistical analysis.
We appreciate the reviewer’s careful reading of the manuscript and figures. In HCC-1954, there has been a mix-up of the annotations ["ns" and "*"] between GW (10 mM) and 1E5 (1 mM). The correct annotations have been put in the revised Figure 1 [GW (10 mM) = p-value (0.047)(“*”), 1E5 (1 mM) = p-value (0.293)(“ns”)]. But in SKBR3, we carried out the statistical analysis again, and exact p-values were obtained as before [GW (5 mM) = p-value (0.097) and GW (10 mM) = p-value (0.070)]. Thus, no changes were included there.
Regarding the discrepancies between MTS assays (absorbance) and cell counting results, we used two types of plates and cell densities in these assays, as mentioned in the method section 2.2 (Lines 92-100). This might result in different sensitivities to treatment and, thus, differences in inhibition patterns. Moreover, the trypan blue exclusion assay directly measures viable cell numbers, while MTS absorbance indirectly measures metabolic activities. Please see the revised Figure 1 and legend in the revised manuscript.
Comments on the Quality of English Language
- Line 17: “HER-positive breast cancer.” Should be “HER2-positive breast cancer.”
The typo was corrected and highlighted in yellow.
- Line 73-75: “Key lipogenesis genes (SREBP1c, ACLY, ACC, FASN, and SCD1) are directly regulated by LXR, and their expression levels upon 1E5 treatment.” This phrase is incomplete.
The corrected sentence can be found on lines 76-78.
- Line 237: “3.2.1. E5 is an LXR Inverse Agonist in HER2-positive Breast Cancer Cells” should be “3.2. 1E5 is an LXR Inverse Agonist in HER2-positive Breast Cancer Cells”
It was corrected and highlighted in yellow on line 240 (numbering changed following other revisions).
We once again appreciate the careful review of the text and for the recommendations which have improved the manuscript.
Round 2
Reviewer 1 Report
Comments and Suggestions for Authors
The manuscript is well revised.
Reviewer 2 Report
Comments and Suggestions for Authors
I appreciate the efforts made by the authors to address the comments and improve the draft. The manuscript has been revised significantly.